Development of a stock trading system based on a neural network using highly volatile stock price patterns

Oh Jangmin jangmin.oh@sungshin.ac.kr
School of AI Convergence, Sungshin Women’s University , Seongbuk-gu , Seoul , South Korea
Moparthi Nageswara Rao
Electronic publication date: 2022 Mar 2
Publication date: 2022
Volume: 8
Electronic Location ID: e915
Received 2021 Dec 21; Accepted 2022 Feb 14
Copyright: ©2022 Oh
Copyright year: 2022
Copyright holder: Oh
License: This is an open access article distributed under the terms of the Creative Commons Attribution License, which permits unrestricted use, distribution, reproduction and adaptation in any medium and for any purpose provided that it is properly attributed. For attribution, the original author(s), title, publication source (PeerJ Computer Science) and either DOI or URL of the article must be cited.
License URL: https://creativecommons.org/licenses/by/4.0/

Keywords: Stock price prediction, Highly volatile stock price pattern, Technical analysis, Neural network

Funding: Sungshin Women’s University Research H20200069 This work was funded by the Sungshin Women’s University Research Grant no. H20200069. The funders had no role in study design, data collection and analysis, decision to publish, or preparation of the manuscript.

==============================
This paper proposes a pattern-based stock trading system using ANN-based deep learning and utilizing the results to analyze and forecast highly volatile stock price patterns. Three highly volatile price patterns containing at least a record of the price hitting the daily ceiling in the recent trading days are defined. The implications of each pattern are briefly analyzed using chart examples. The training of the neural network was conducted with stock data filtered in three patterns and trading signals were generated using the prediction results of those neural networks. Using data from the KOSPI and KOSDAQ markets, It was found that that the proposed pattern-based trading system can achieve better trading performances than domestic and overseas stock indices. The significance of this study is the development of a stock price prediction model that exceeds the market index to help overcome the continued freezing of interest rates in Korea. Also, the results of this study can help investors who fail to invest in stocks due to the information gap.

Introduction

Predicting stock prices has long been of interest to many related fields including economics, mathematics, physics, and computer science. There is an ongoing debate about whether or not it is possible to predict stock prices and, if it is possible, how much these predictions can outperform the market. However, the field of AI (artificial intelligence) has recently reported price forecasting techniques with the application of various machine learning techniques that show a significant level of statistical confidence (Hsu, 2011; Hadavandi, Shavandi & Ghanbari, 2010; Armano, Marchesi & Murru, 2005; Ding et al., 2015; Jiang, 2021). A number of studies building intelligent trading systems have also been conducted based on the results of these AI price forecasting techniques (Lin, Yang & Song, 2011; O et al., 2004; Song, Lee & Lee, 2020).

O et al. (2004) and O et al. (2006) concluded that trading performance can be additionally improved by training and utilizing independent predictors for different stock price patterns.

Most of the existing stock price prediction technical analyses have based the input features on the moving average (MA) stock price, which can effectively express the recent trends of price fluctuations. For example, the MACD (moving average convergence divergence) utilizes the difference between the long term and short-term moving average to represent the convergence and divergence of the moving average values.

O et al. (2006) performed pattern-defined predictions using patterns related to a crossover, reversal into an uptrend, and reversal into a downtrend among 5-day, 10-day and 20-day moving average lines.

However, all of the MA-based technical indicators, including the MACD, have a ‘time-lag’ limitation because buy and sell signals are mostly generated after price trends have already been developed. This study will attempt to predict highly volatile stock price patterns by introducing the concept of the ‘upper limit price’, which is defined independently from the moving average. This study will also utilize Japanese candlestick indicators and more short-term technical indicators to complement the time-lag problem.

In the Japanese candlestick indicator, a candlestick summarizes the intraday variations of a stock price, expressing the differences between the opening price, highest, lowest, and closing prices, through which the most recent price fluctuations can be summarized more closely.

According to various empirical analyses of the Korean stock market, the Korean stock market shows market inefficiency due to information asymmetry (Lee, 2007; Bark, 1991). Although market inefficiency is lower than that of larger foreign stock markets, there is still an issue of the information gap. Therefore, this paper suggest that special investment and analysis information to overcome market inefficiency. However, since technical analysis indicators are price- and chart-based information that many people already know, a new chart analysis technique is needed.

The efficient market hypothesis asserted by Fama (1965) is rejected if it exceeds the market rate of return using specific information. This study proposes a new ‘highly volatile stock price pattern’ that does not yet exist in technical analysis. Using the pattern proposed in this paper, it is possible to develop a predictive model that exceeds the market return. The results of this study are in conflict with the efficient market hypothesis.

In summary, this study assumes market inefficiency in the Korean stock market and provides new information that is expected to affect price fluctuations. The highly volatile stock price pattern is defined by the relationship between the ‘upper limit’ and stock price in the Korean stock market. Through fund simulation it was found that investors can obtain efficient returns through a deep learning stock price prediction model using highly volatile patterns.

It was also found that it was difficult to predict stock prices by only analyzing simple charts such as moving averages, so a definition of a particular pattern of variation is needed. This pattern can be found when there are stock prices that show an upper limit. This pattern can also appear over various periods of time even when the chart shows the characteristics of a random walk.

The experiment was conducted based on related research that the Korean stock market shows more trend changes than random work characteristics (Aggarwal, 2018; Ryoo & Smith, 2002; Ayadi & Pyun, 1994).

Studies related to stock market analysis from China, India, and Mongolia, which are similar to Korea, were also unable to prove the random walk theory according to the stock market (Han, Wang & Xu, 2019; Damdindorj, Rhee & Choi, 2016).

The present study will show that the proposed stock trading system can achieve better trading performances than domestic and overseas stock indices by performing pattern-specific training on highly volatile stock price patterns. The experiments were conducted on the stock variation data extracted from the price data of approximately 2,000 stocks listed in the KOSPI and KOSDAQ markets. The limitations of existing studies and the strengths of this study are as follows.

Limitations of existing studies

1. Limitations of Profitability Verification: Existing studies suggest only accuracy or prediction error using stock price prediction models. However, the actual performance of the stock price prediction model requires return verification.

2. Limitations of using simple input features: Existing studies have used very simple input features such as simple price features, volume, price rate of change and so on. Simple input features are limited and using more advanced input features can greatly influence the performance of predictive models.

3. Limitations on prediction accuracy: Most of the predictive models in existing studies have not achieved high accuracy.

Strengths of this study

This study conducted a clear performance verification through yield comparison with several indices. In addition, although a simple deep neural network was used, advanced input features improved the performance of the stock price prediction model. Although there are time series models such as LSTM, this study uses high-volatile pattern filtering input features. As a result, stock price data of a portion without the corresponding pattern is partially filtered. Therefore, the continuity of time series data is insignificant. As a result, deep learning helped us identify the importance of input features such as high volatility patterns in predicting stock price. The deep neural network we used is a rather simple structure, but its sufficient performance has been verified. The novelty of this study is summarized as follows.

• First, an advanced filtering technology that can improve the performance of the stock price prediction model was proposed. The pattern of stock price fluctuations proposed in this paper was created based on the results of analyzing domestic stock charts. Since it is a pattern generated based on the actual chart, it reflects the market well and shows excellent performance compared to the existing moving average-based pattern.

• Second, it enables investors to make comfortable investments without daily data analysis. By using the neural network model, a lot of data can be handled at once, and the prediction results are reliable results that have been verified through a sufficient period of fund simulation.

• Third, advanced filtering technology enables sufficient stock price prediction even in a simple deep learning model. Most of the existing studies focus on the structure of the model when conducting deep learning stock price prediction studies. For example, performance is compared using multiple models for the same data. However, in addition to the issue of selecting the structure of the model, the composition of data and the importance of filtering algorithms are proposed through this study.

This paper is organized as follows: ‘Related Works’ describes related studies; ‘Background’ describes the knowledge used prior to the main methodology of this study; in ‘Materials & Methods’, the moving average-based patterns typically utilized in existing studies will be introduced, the formal definition of the three highly volatile stock price patterns will be presented, and the meanings of the individual patterns will be described through chart demonstrations; in ‘Experiments’, the input features and the target value of the neural network learning system will be described; in ‘Results of experiments’, the experimental results will be presented; and, finally, future research directions will be suggested in the ‘Conclusion’.

Related works

Researches related to stock price prediction have traditionally been conducted using ARIMA (Benvenuto et al., 2019; Ariyo, Adewumi & Ayo, 2014), Regression (Refenes, Zapranis & Francis, 1994; Yang, Chan & King, 2002), and Bayesian (Pella & Masuda, 2001) to reflect the characteristics of time series data.

ARIMA is a statistical model widely used in the financial sector. However, it is a model that is used exclusively for short-term predictions, and has the disadvantage that it is difficult to confirm long-term investment performance. Since stock price prediction is closely related to profits, direct investment in a short-term verified model can lead to risks. In addition, for the reason that the amount of stock data accumulated from the past is very vast, a model that can handle large amounts of data is needed. In this regard, there is also a study result that predicts the index using the ARIMA model has an accuracy of up to 38% (Devi, Sundar & Alli, 2013). However, the prediction accuracy was very low to carry out actual investments using this model.

Baysian is a model that can perform classification based on probabilistic theory and used to predict stock prices in the past. However, with the recent development of artificial neural networks, it is widely used as a comparative model. Baysian is also evaluated as not suitable for mid or long-term prediction, such as the ARIMA model. In related studies using the Bayesian model, predictions were performed with up to 78% accuracy (Malagrino, Roman & Monteiro, 2018). This was better performance than the ARIMA model, but it showed lower values than the neural network model to be described below.

Above all, these existing predictive models often misinterpret information due to underfitting and overfitting problems, so they often do not help much in decision-making activities for stock price prediction. In addition, it has already been proven that neural networks perform better than traditional methods (Bustos & Pomares-Quimbaya, 2020). For the above reasons, neural network-based technique has been used a lot in recent stock price prediction.

Representatively, there is a study using Elliott Wave Indicator as a stock price prediction study using neural networks (Lakshminarayanan et al., 2006). This study used data based on technical analysis, and the model accuracy was 93.83%. This study used its own technical indicators and made predictions for five stocks. Based on these findings, it is possible to assume that more advanced input features can lead to improved model performance.

Several other papers similar to this paper used machine learning techniques such as regression and SVM, models such as CNN and LSTM, or ensemble techniques to predict stock prices (Oncharoen & Vateekul, 2018; Liu & Wang, 2018; Jiang, 2021).

Cao & Wang (2019) attempted to predict the stock index of various countries using the CNN application model. Only historical data was used as input features, and the down-sampling and convolution techniques belonging to CNN were mainly used to improve stock price prediction performance. As a result, they suggested that the CNN-SVM mixed model had the best performance. However, simple input features were used, and the exact performance and return of the model were not verified, so it was difficult to confirm the actual performance of the CNN-SVM model.

Another study (Guang, Xiaojie & Ruifan, 2019) looked at the rate of the return of the stock price forecast model. This model is not a period-specific return, but an absolute return that does not take into account investments and assets that may vary from person to person. In addition, the profits earned have no comparison with the stock index, which can be regarded as a rate of return due to market rise.

A similar study (Pang et al., 2020) included word-embedding techniques such as LSTM (Long Short-Term Memory). Here stock data, which is a time series characteristic, is referred to as a stock vector, and is used as an input feature. The model they developed tried to predict the Shanghai Stock Index and showed an accuracy of about 57%, but the study did not have a comparison of returns through prediction. High volatile stock price prediction model derived higher accuracy than the model presented by Pang et al. In addition, only a few studies (Feng et al., 2018; Araújo et al., 2019) using various in-depth models have both derived returns and evaluated their performance, and most studies cannot guarantee a clear return because they provide no comparison with the stock index.

Recent evidence suggests that input feature selection is very important in model learning. The selection of key input features can lead to improved (Hooshmand & Gad, 2020). Accordingly, in this paper, the study was conducted with a focus on data composition and novelty rather than the structure of the model.

Background

MA (Moving average)-based patterns

Before examining the high volatile stock price pattern using the Japanese candlestick indicator, which is the focus in this study, the meaning and limitations of the four MA-based patterns presented in related studies are examined, taking the ‘divergence’ pattern and ‘reversal’ pattern of MA (Moving Average) as examples. O et al. (2006) that defined and used the patterns based on the moving average. The moving average is the average stock price over a period of time and is used to summarize stock price trends. Moving average is also denoted by MA in which 5, 10, 20 days and so on are used as the window of time; as an example, the 5-day moving average of stock s at trading day tcan be calculated as follows: (1) MA5ts=15∑k=04closet−ks.

In the same way as above, the volume can also be calculated as the moving average of the volume, and the 5-day volume moving average of stock trading day can be counted as follows: (2) VMA5ts=15∑k=04volumet−ks

where closets is the closing price of the trading day t. The slope of the line connecting a moving average to another moving average is denoted as Grad and can be calculated using the following Eq. (3): (3) Grad5ts=MA5ts−MA5t−1sMA5ts.

Equation (4) defines the training target set Dbear that corresponds to the divergence pattern. Divergence refers to when the short-term moving average is located relatively below the longer-term moving averages, resulting from the continuation of the downward trend of the stock price for a considerable period of time. Equation (4) represents that the 5-day moving average is smaller than the 10-day moving average and the 10-day moving average is smaller than the 20-day moving average: (4) Dbear=xs,t,os,t|MA5ts<MA10ts<MA20ts,s∈α,t∈β

where xs,t is a vector of the input feature, and os,t is the target value representing the price fluctuation after the occurrence of the pattern; α and β are the entire stock set and the entire trading day set, respectively. Figure 1 is an example of the charts corresponding to a divergence pattern. The description of the graph in Fig. 1 is as follows. Sixteen trading days from A to B correspond to the divergence pattern. In this case, relatively steep price rises were shown around point B; but, in general, even if a rebound was to appear, the rise would not be that large.

Figure 1 MA divergence pattern.

Equation (5) defines the training target set DTU that corresponds to the reversal to uptrend pattern. Reversal to uptrend means that one of the moving average lines reverses from a downtrend to an uptrend; trading day A, B, C in Fig. 2 show reversal to uptrend patterns. Among these, A shows the case where the 5-day MA line reversed to an uptrend due to the sharp rise of the stock price.

Figure 2 MA reversal to uptrend pattern.

This case shows the weakness of MA as a prediction indicator because the price had already risen before the trading signal was issued; the phenomenon of generating the signals after the price movement has already occurred is called ‘time-lag’. This MA-based pattern arises very frequently so it has the strength of using a large amount of training data, but time lag decreases this pattern’s predictive ability. (5) DTU=xs,t,os,t|Grad5t−1s<0&&Grad5ts>0||Grad10t−1s<0&&Grad10ts>0,||Grad20t−1s<0&&Grad20ts>0,s∈α,t∈β.

Materials & Methods

Highly volatile stock price patterns

This section describes a technique for filtering data showing a high volatile stock price pattern on a stock chart. There are a total of three filtering algorithms, and using them, data with high fluctuation patterns form a cluster. Machine learning-based algorithms such as k-means were not necessarily used when creating clusters. This is because the time it takes to create a cluster is very short compared to similar studies (Alguliyev, Aliguliyev & Sukhostat, 2019).

In the pattern-based stock trading system, multiple independent predictors are trained on the data clustered in line with the stock price patterns and employed in the final trading. In this paper, the highly volatile stock price patterns will be defined and employed as a way to achieve more predictability as an extension of pattern-based prediction techniques. Korean stock markets set the legal limits of price fluctuations in a day; both the KOSPI and KOSDAQ markets apply ±15% of the previous day’s closing price to the price fluctuation limits. In general, the ‘upper limit’ refers to when the closing price of a particular trading day is closest to a 15% rise from the previous day’s closing price, or about a 14% rise in the price depending on the price band of the item. In Eq. (6), Sanghants represents the closing price at the upper limit of stock s on trading day t and is defined as follows: (6) Sanghants=trueifclosets≥1.14×closet−1sfasleotherwise

where closets refers to the closing price. The three highly volatile stock price patterns are defined based on the definition of the upper limit. If the price of a particular stock has risen to the upper limit, the price volatility of the subsequent trading days is bound to be expanded. The rise of the price to the upper limit should be accompanied by large transaction volume because a short-term surge and plunge may occur due to the collective psychology of the trading public. The highly volatile stock price patterns in this paper deal only with the case where price adjustments were made 1 to 2 trading days after the upper limit price appeared. The time target used for predictions was when both the first rising wave, represented by the rising to the upper limit price, and the first falling wave, represented by the adjustment, have been completed so extreme volatilities have been relaxed. High volatile stock price patterns were found by chart analysis experts who analyzed charts over the years. This pattern is actually mainly seen in mid- to low-priced stocks in the Korean stock market. Investors have to look directly at a vast amount of data to utilize this pattern for real investment. However, if the prediction model using deep learning is properly defined, it is easy to predict when the price rises after a certain pattern, and even check whether it can actually make profits.

Adjustment pattern with one candlestick after consecutive upper limits (p1)

An adjustment pattern with 1 candlestick after consecutive upper limits is when the first rising wave is so strong that the upper limit prices appear consecutively; it can be calculated using the equation below (7). opentsrepresents the open price of stock s on trading day t. Figure 3 shows an example of p1 where (a) is a normal example in which a p1pattern occurs and the upper limit price appears next day, and (b) is a counter example where the empty rectangular, called a negative candlestick, represents when the closing price is lower than the opening price, meaning that the price became lower after intraday trading. The positive candlestick filled with the grey color is the opposite. The last condition of the condition statement means that the candlestick in the pattern formation for the day should be a negative candlestick or should be a positive candlestick of less than 5% of the difference between the opening price and the closing price. If the positive candlestick is larger by a difference of more than 5%, the price is considered adjusted. (7) p1ts=trueifSanghant−2s=trueandSanghant−1s=trueandclosets−opentsclosets<0.05falseotherwise.

Figure 3 (A–B) A normal example and counter example of the p1 pattern.

Adjustment pattern with one candlestick after single upper limit (p2)

An adjustment pattern with one candlestick after a single upper limit is when the upper limit price condition is replaced by the single upper limit price; Fig. 4 shows a normal example and a counter example of the case. Due to the relatively weaker rising intensity, it is estimated that the ratio of normal examples is likely to be rather lower than p1 patterns. (8) p2ts=trueifSanghant−2s=falseandSanghant−1s=trueandclosets−opentsclosets<0.05falseotherwise.

Figure 4 (A–B) A normal example and counter example of the p2 pattern.

Adjustment pattern with two candlesticks after upper limits (p3)

An adjustment pattern with two candlesticks after the upper limits is the case in which the price has been adjusted over two trading days after the upper limit price, or when a negative candlestick or a small positive candlestick (less than 5% in size) forms after p1 or p2 is formed. Figure 5 shows the examples of p3. The last candlestick on each chart of (a) and (b) is the price fluctuation immediately after the pattern occurs. (9) p3ts=trueifSanghant−1s=trueandcloset−1s−opent−1scloset−1s<0.05andclosets−opentsclosets<0.05falseotherwise.

Figure 5 (A-B) A normal example and counter example of the p3 pattern.

These examples of the three patterns examined above imply that price fluctuations after the pattern occurs can vary depending on the slope of the moving average and the form of the candlestick. As an example, the under tail is attached to the last candlestick in all the patterns shown in the three normal examples. This means that the stock ended with the emergence of buying powers leading the rebound in price after trading hours and is more likely to be bullish the next day. Since the various factors act in combination on the direction of the stock price after the appearance of the pattern, the neural network training that will be described in the next section is needed to utilize these patterns in the trading system.

Experiments

Input features configuration for neural network

In order to train the neural networks for future price predictions for each pattern presented in the previous sections, the input feature set constructing an input vector xs,t was used for the input to the neural network from the training data and the target value corresponding to the desired output was defined. Disparity, representing the distance between MA and the current price, is denoted as Disp and the Disp from the 5-day MA line can be calculated using the following equation: (10) Disp5ts=closets−MA5t−1sMA5ts.

Apart from the moving average line, the input features relating to Japanese candlesticks include: RC (rate of change) in the trading day price compared to the previous day, Body, US (upper shadow), and LS (lower shadow). These are defined in Eqs. (11) through (14), respectively: (11) RCts=100×closets−closet−1sclosets

(12) Bodyts=100×opents−closet−1s minopents,closets

(13) USts=100×hights− maxopents,closetsmaxopents,closets

(14) LSts=100×minopents,closets−lowts minopents,closets

where opents,hightsandlowts are the opening, highest, and lowest price of the trading day t. The input vector xs,t of each predictor, including these indicators, is as follows: (15) xs,t=RCts,RC t−1s,RC t−2s,Body ts,Body t−1s,Body t−2s,US ts,LS ts,Grad5 ts,Grad10 ts,Grad20 ts,Disp5 ts,Disp10 ts,Disp20 ts,V Grad5 ts,V Grad10 ts,V Grad20 ts,V Disp5 ts,V Disp10 ts,V Disp20 ts

where V Grad is the slope of the volume moving average line. V Disp is the difference between the volume moving average and the total volume; these two indicators can be calculated by entering the total volume instead of the close price in the equations of the Grad and Disp. Each input feature should be normalized as a value between 0 and 1 before used.

Experimental process and environment

The first of the stock price prediction process takes data from the stock database for a certain period of time. Thereafter, the data of the part showing the high volatile pattern is filtered and the input feature is calculated. The calculated result is saved as a text file and is used for deep learning. The text file consists of training, validation, testing, and fund simulation files, and finally the model outputs dates and stocks that are expected to rise by more than 10%.

The stock database stores daily KOSPI/KOSDAQ data from October 1990 and it updates data every day. Therefore, even if the composition of the stock index changes, the changed information is newly added to the database, so it is possible to predict the changed index.

A prediction model using the neural network structures as shown in Fig. 6 below was trained using 20 input features and binary classified target vectors. The target vector uses a binary classification format, and if the price rises by more than 10% within 5 days, it will be marked [0,1] otherwise [1,0].

Figure 6 Neural network structure and target vector configuration.

Experiments will be conducted on a desktop with 18.04 versions of Ubuntu with RTX 3070 9GB graphics card. The model was constructed using Keras in Tensorflow 2.0 (Abadi et al., 2016) and the hidden layer was composed of three hidden layers. In addition, the use of detailed parameters is as follows. Tanh was used as an activation function for each layer, and the learning rate was set to 0.01. In addition, the dropout ratio was set to 0.5.

Experiment results

Performance evaluation of prediction model

This section will present the experiment results of applying the proposed trading system to the prices of 2,268 stocks listed on the KOSDAQ and KOSPI markets of the Korean Stock Exchange. The entire dataset used in the experiment is divided into 4 sub-sets; the details are shown in Table 1. In other related studies, there are no results of measuring returns through the prediction model. In studies related to stock price prediction, not only the accuracy of the prediction model but also the measurement of the rate of profit should be conducted at the same time. In this paper, a fund simulation dataset is additionally configured for this purpose. Training, verification, and test data are used exclusively for determining prediction models. Only the data that was not used to generate the prediction model was used for fund simulation. This process is for cross-validation and accurate rate of profit measurement.

Table 1 Data set for building the trading system.

Name of dataset	Period	
Training set	October 2017–June 2019	
Validation set	July 2019–December 2019	
Test set	January 2020–April 2020	
Fund simulation set	May 2020–December 2020	

This study was aimed only at predicting the domestic stock market, so only data from the KOSPI and KOSDAQ were used. . The total number of KOSPI and KOSDAQ stocks in Korea is about 2,436. In this study, data were collected from October 2017 to December 2020 based on KOSPI and KOSDAQ stocks, the total number of data used for training, verification, and testing, and additional fund simulations exceeded 2 million. Because of the large amount of data, other analytical techniques and theories were needed to add data from overseas stock markets.

The experiments showed that as training progressed, loss decreased and accuracy increased, as shown in Fig. 7. The loss calculation used MSE, and the equation is as follows. Where N is the number of samples, P is the predicted value and A is real value. Mean square error is defined as the variance between predicted and actual values (Namasudra, Dhamodharavadhani & Rathipriya, 2021). (16) MSE=1N∑1NPi−Ai2.

Figure 7 Training and test loss and accuracy graph.

Test datasets showed a slight increase in loss and a slight decrease in accuracy with each epoch. However, we can see significantly better numerical results than the training process.

The evaluation of the prediction model is shown in Table 2. The model’s evaluation was performed on two criteria: accuracy and F1 score. Accuracy is a metric for the classification model as a percentage of the total predictions performed. The F1 score is the harmonic mean of precision and recall (Chakraborty et al., 2020). The equation for accuracy and f1 score is as follows.

Table 2 Model evaluation results.

	Accuracy	Precision	Recall	F1-Score	
Model evaluation Results	0.9623	0.9638	0.9638	0.9638	

(17) Accuracy=100×TruePositives+TrueNegativesTruePositives+TrueNegatives+FalsePositive+FalseNegatives

(18) F1score=2∗Precision∗RecallPrecision+Recall

where:

Precision=True PositivesTrue Positives+False Positive,Recall=True PositivesTrue Positives+False Negatives.

The experiments showed that the stock price prediction model using highly volatile stock price patterns finally showed an accuracy of 96.23% and an F1 score of 0.9638. This result is slightly lower or better compared to other classification models (Agrawal et al., 2021; Ndichu, Kim & Ozawa, 2020) not stock price prediction models. However, in the case of stock price forecasts, this figure can be said to be a good result due to high uncertain volatility.

The predictors by pattern were constructed using the training data by pattern, and the optimum trading policies were selected by performing the integrated multiple simulation presented in Lee (2007), applying the ‘trading policy selection set’ to each predictor. Here, the integrated multiple simulations refer to the technique to find the optimal trading policies best suited to a given predictive neural network. For example, when a prediction is performed on the fund simulation set, stocks and dates that will rise more than 10% are derived. The stocks to rise consist of those with a neural network threshold of more than 0.5. It was found that the optimum trading policy had a 20% in profit realization rate, −12% in stop loss rate, and a holding period of 19 days.

The results of this experiment were compared with similar studies using other filtering algorithms to derive the results shown in Table 3. Highly volatile filtering algorithm defines the pattern of fluctuations in stock prices using the concept of upper limits. In comparison, the remaining three algorithms are ‘Resisted plunge filtering’, ‘Nosedive filtering’, and ‘Rise stock filtering’, respectively. ‘Resisted plunge’ refers to the type in which an ascending stock drops for a short period. ‘Nosedive’ literally means a slump. In this case, it is to collect stocks that have shown a period of collapse. ‘Rise stock’ filtering represented the long-term upward trend (Song & Lee, 2018).

Table 3 Comparison of the accuracy between the proposed scheme and other stock filtering schemes.

	Resisted plunge filtering	Nosedive filtering	Rise stock filtering	Our Model with highly volatile stock price patterns filtering	
Accuracy	72.39%	75.11%	64.93%	96.23%	

Experiments were conducted using the same data and model structure. The experimental results showed that Resisted plunge filtering achieved 72.39% accuracy, Nosedive filtering achieved 75.11% accuracy, and Rise filtering achieved 64.93% accuracy. In contrast, it can be seen that the highly volatile pattern filtering algorithm is 96.23%, which is higher than the accuracy of the other filtering algorithm.

Results of fund simulation

As a result of the fund simulation using optimal trading policies, it was able to earn better profits than the domestic stock index during the same period. Industrial indicators were not predicted separately because they were both included in the KOSPI and KOSDAQ already as this data contains all stock data by industry. As a consequence, the number of stock trades was small because the stock price pattern showing a highly volatile pattern did not appear much in the fluctuation pattern of all stocks. However, when a highly volatile pattern occurs, It was found that there is a high probability that a subsequent upward pattern will appear. In conclusion, highly volatile stock price patterns play a big role in predicting stock price rise.

Additionally, to supplement the stock price prediction performance of the high volatile model proposed in this paper, fund simulation results were compared it with the Nikkei 225, NYSE, and NASDAQ, which are representative stock indices in Japan and the United States. The results are shown in a graph with the domestic stock index in Fig. 8. Even when other indices were lowered, high volatile stock price prediction model showed a steady upward graph. Finally, this model can earn the highest cumulative return. Figure 8 shows the percentage of returns from each asset.

Figure 8 Comparison of the cumulative profit rate between domestic and overseas stock indices and the prediction model.

The reasons that the trading system proposed in the paper achieved better trading performance than the domestic and overseas stock indices are: first, including the microscopic price change processes of the most recent three days in the training input features helped train the neural network training; second, defining the scope of each pattern by using more strengthened constraints than the moving average pattern seems to have contributed to the improvement of learning performance as well as the ultimate trading performance.

Conclusions

This paper constructed a pattern-based stock trading system which learned data corresponding to the three highly volatile stock price patterns and utilized that data for trading. The highly volatile stock price pattern can be observed over a long period of time and almost guarantees a short-term rise after the pattern occurs.

The significance of this study is the development of a stock price prediction model that exceeds market indices to overcome the continued freezing of interest rates in Korea, Japan, and the US Also, the results of this study can help investors who fail to invest in stocks due to the information gap. If special analysis techniques and indicators such as high volatility patterns are proven to be effective through this research method, individual investors can use these methods in the future. In addition, a number of other patterns of variation can be added to expand the model, and if a positive return is proven, anyone can use the fund simulation for their own investments.

Additional studies will have to be conducted to achieve much better trading performances through microscopic analysis and classification of other highly volatile stock price patterns not used in this paper. Improving the input feature set used in this paper, and reflecting the variations of the periods in the target values may also help achieve better results. In addition, performance changes will be measured by applying the NAR Neural Network Time Series (NAR-NNTS), a recently studied model (Namasudra, Dhamodharavadhani & Rathipriya, 2021) that is suitable for data with uncertainty in future studies.

Finally, this study was conducted only using Korean stock data as the Korean stock market is very different from overseas stock markets such as those in the United States. However, in order to highlight the strengths of prediction model by performing cross-comparison with overseas stock indices, comparisons with three indices were added. Price restrictions such as upper and lower limits are usually used in Asia including China, Japan, and Thailand. This paper identified patterns inspired by Japanese candle charts and compared them with the Japanese stock index. In addition, it completed the yield comparison by adding comparison with overseas stock markets such as the US As a result, the proposed model showed a higher return than the overseas market growth rate during the same period.

Additional Information and Declarations

Competing Interests

Author Contributions

Data Availability

The authors declare there are no competing interests.

Jangmin Oh conceived and designed the experiments, performed the experiments, analyzed the data, performed the computation work, prepared figures and/or tables, authored or reviewed drafts of the paper, and approved the final draft.

The following information was supplied regarding data availability:

The data and source codes are available at GitHub: https://github.com/chfhdahwk/Hivol_Stock_Prediction.git.

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
