# Peer review of "Development of a stock trading system based on a neural network using highly volatile stock price patterns"

_PeerJ Computer Science, doi:10.7717/peerj-cs.915_

## Round 0.1 · original submission · Minor Revisions

Good work, however, please update your manuscript according to the reviewers' suggestions.

Reviewer 1 ·

Basic reporting

This paper proposes a pattern-based stock trading system using ANN-based deep learning and utilizing the results to analyze and forecast highly volatile stock price patterns. Three highly volatile price patterns containing at least a record of the price hitting the daily ceiling in the recent trading days are defined. The implications of each pattern are briefly analyzed using chart examples.

Here, the training of the neural network has been conducted with stock data filtered in three patterns and trading signals were generated using the prediction results of those neural networks. Using data from the KOSPI and KOSDAQ markets, this paper shows that the proposed pattern-based trading system can achieve better trading performances than domestic and overseas stock indices.

Experimental design

1. How the experimental environment has been developed?
2. Why the dataset is divided into four parts?

Validity of the findings

1. How the accuracy is 96.23%?
2. The proposed scheme must be compared with at least two existing schemes.
3. There must be a discussion on how the results are generated. Technical details are missing.

Additional comments

1. Motivations of the paper are not clear.
2. Contributions must be represented point-wise.
3. In the "Related Works" section, the existing schemes must be discussed one by one. Delete the limitations from this section and add them in the Introduction section.
4. How the price pattern is chosen.
5. The proposed scheme is unstructured. divide the proposed scheme in many sub-sections, and then, discuss the entire proposed scheme under each sub-section.
6. Equations and figures are not represented properly. All the key terms must be defined.
7. It is hard to identify the novelty of the proposed work.
8. The English language is very poor. Never use I, we, or our in a Research Article.
9. The organization of the paper is poor. Add section number.
10. Important references are missing. Add the following references:
“Efficient algorithm for big data clustering on single machine”, CAAI Transactions on Intelligence Technology, vol. 5, no. 1, pp. 9-14, 2020.
“Ensemble algorithm using transfer learning for sheep breed classification”, Proc. of the 2021 IEEE 15th International Symposium on Applied Computational Intelligence and Informatics (SACI), IEEE, Timisoara, Romania, pp. 199-204, 19-21 May 2021.
“Enhanced neural network based univariate time series forecasting model”, Distributed and Parallel Databases, 2021. DOI: 10.1007/s10619-021-07364-9
“Deobfuscation, unpacking, and decoding of obfuscated malicious JavaScript for machine learning models detection performance improvement”, CAAI Transactions on Intelligence Technology, vol. 5, no. 3, pp. 184-192, 2020.
“Fast and secure data accessing by using DNA computing for the cloud environment”, IEEE Transactions on Services Computing, 2020. DOI: 10.1109/TSC.2020.3046471
“Feature selection approach using ensemble learning for network anomaly detection”, CAAI Transactions on Intelligent Technology, vol. 5, no. 4, pp. 283-293, 2020
“Securing multimedia by using DNA based encryption in the cloud computing environment”, ACM Transactions on Multimedia Computing, Communications, and Applications, vol. 16, no. 3s, 2020. DOI: https://doi.org/10.1145/3392665
“Nonlinear neural network based forecasting model for predicting COVID-19 cases”, Neural Processing Letters, 2021. DOI: 10.1007/s11063-021-10495-w
“IFODPSO-based multi-level image segmentation scheme aided with Masi entropy”, Journal of Ambient Intelligence and Humanized Computing, vol. 12, pp. 7793-7811, 2021. DOI: https://doi.org/10.1007/s12652-020-02506-w

Reviewer 2 ·

Basic reporting

The author didn’t properly introduce the problem correctly. There is lot of Finance related technical information that’s presented to the reader from the beginning without properly clarifying what it is. Author should work on easing the reader into the topic.



The quality of the figures presented are not publishing ready and should be redone as a high quality image

Experimental design

Each stock market index uses proprietary methods to determine which companies or investments to include and that can change over time. How is that handled here?

Validity of the findings

This is a time series prediction problem, why are we not using a variation of LSTM’s? LSTM’s almost always guarantee better results than DNN. Authors should experiment with modern methods rather than leaving it at DNN

·

Basic reporting

Impact of volatility on the assets prices changes play a crucial for traders/investors for capturing positive returns on their investments and with the advancement of technology revolution crept into the stock markets arena. Since the turn of the century financial markets have undergone two major changes, (a) massive information (b) speed of strategy execution. The paper under the review speaks of the same tone for picking up a strategy and to discover a price amidst of volatility. But the paper need to speak conceptually more on drawback in the current existing patters while proposing for the new trading mechanism.

Experimental design

1. Provide the complete implementation process?
2. Does any specific reason for datasets is divided into more parts?

Validity of the findings

1. How was 96.23% accuracy arrived at, what was the measuring method used to arrive at 96.23% accuracy?
2. The proposed scheme must be compared with at least ‘four’ traditional methods.
3. Need to add more technical novelty details.

Additional comments

1. Motivation of the paper was not clear.
2. Need more reference on related work, on the other hand with what ever reference works were quoted the authors haven’t touched up on criticism/drawback/limitations in the related works.
3. Research work is silent in recommending a particular method/process for measuring accuracy.

---

## Round 0.2 · accepted · Accept

Good work! You have addressed all reviewers' concerns.

Reviewer 1 ·

Basic reporting

This paper proposes a pattern-based stock trading system using ANN-based deep learning
and utilizing the results to analyze and forecast highly volatile stock price patterns.

Experimental design

The experimental design is convincing.

Validity of the findings

The results show the efficiency of the proposed scheme.

Additional comments

The paper can be accepted.

·

Basic reporting

The author has been clear in presenting his thoughts and the literature reference are sufficient .

Experimental design

1. Author have provided the complete implementation process;
2. The author has rectified the specific reason for division of the datasets.

Validity of the findings

1.Validation are presented much better way in substantiating the argument.
2. They had clearly compared with few methods that added novelty to their approach.

Additional comments

Motivation for taking up the paper and summering the paper are clear.